# The Effect of Free Weight Resistance Training on Cognitive Function Explored Through Eye Tracking: A Randomized Double-Blind Clinical Trial

**DOI:** 10.3390/bs15010077

**Published:** 2025-01-17

**Authors:** Cristián Mateluna-Núñez, Romualdo Ibáñez-Orellana, César Campos-Rojas, Andrea Santana-Covarrubias, Rodrigo Fuentes Figueroa, Ricardo Martínez-Flores

**Affiliations:** 1Escuela de Educación Física, Pontificia Universidad Católica de Valparaíso, Valparaíso 2340025, Chile; rodrigoafuentesfigueroa@gmail.com; 2Instituto de Literatura y Ciencias del Lenguaje, Pontificia Universidad Católica de Valparaíso, Valparaíso 2340025, Chile; romualdo.ibanez@pucv.cl (R.I.-O.); cesar.campos@pucv.cl (C.C.-R.); andrea.santana@pucv.cl (A.S.-C.); 3Instituto de Literatura y Ciencias del Lenguaje, Pontificia Universidad Católica de Valparaíso, Millennium Nucleus for the Science of Learning (MiNSoL), Valparaíso 2340025, Chile; 4IRyS Group, Escuela de Educación Física, Pontificia Universidad Católica de Valparaíso, Valparaíso 2340025, Chile; ricardo.antonio.martinezf@gmail.com

**Keywords:** cognition, resistance training, cognitive function, eye tracking, text processing, exercise, executive function

## Abstract

In this study, we investigated the impact of a 10-week free weight resistance training (RT) program on cognitive function in healthy young adults. In this randomized controlled trial, 18 participants were assigned to either an experimental or control group. We assessed cognitive function by using eye-tracking (ET) technology during text processing tasks. First-pass reading times (FPRTs) and total reading times (TRTs) were measured. Results revealed a significant three-way interaction between group, moment, and syntactic complexity in FPRTs, demonstrating training effects on cognitive processing. The experimental group showed a distinctive shift in processing patterns: from longer times in low complexity pre-intervention to increased times in high complexity post-intervention, particularly in early processing measures (FPRTs). Complementary analyses of strength improvements showed that increased strength was associated with enhanced attention allocation to complex structures and improved processing efficiency for simpler texts, suggesting RT’s potential to modulate cognitive function.

## 1. Introduction

Neuromuscular function is one of the most important and determining factors in athletic performance ([17]; [39]) and a key objective of resistance training (RT), such as weightlifting; however, RT, defined as a physical exercise program designed to maintain and improve muscular strength, endurance, and lean muscle mass ([43]; [37]), is also vital for maintaining health and enhancing functional capacity in both young and older populations ([29]; [26]). In this regard, RT has been associated with a reduction in all-cause mortality, decreased cardiovascular risk and blood pressure ([19]), improvements in glucose metabolism ([15]), benefits to bone, tendon health, and cartilage loss prevention ([38]), cancer treatment ([30]), the prevention of muscle loss, and reductions in obesity by promoting fat loss ([49]). Alongside physical benefits, there is also evidence suggesting that RT can prevent cognitive and neuronal declines ([11]; [36]). The physiological explanation for this cognitive improvement considers the reduction in inflammation ([12]) and the increase in blood flow to the brain ([2]; [50]). Another reason is that RT enhances the availability of various growth factors on the brain, such as insulin-like growth factor-1 (IGF-1) ([19]) and vascular endothelial growth factor (VEGF) ([11]), which regulate exercise-induced angiogenesis and neurogenesis in the hippocampus. RT also increases the production of brain-derived neurotrophic factor (BDNF), which is involved in neuroplasticity and learning ([18]).

While the existing literature presents encouraging potential regarding the use of RT as a tool to enhance cognitive function, it also reveals specific aspects within the research domain that warrant further exploration. These aspects include populations, programming parameters, and outcome measures. Most studies have concentrated on populations experiencing age-related or disease-related cognitive decline ([11]), but there are relatively few studies that have specifically targeted healthy young adults and demonstrated a positive impact of RT ([7]; [10]). Regarding programming parameters, existing studies have mainly used exercises with machine weights, or a combination of machine and free weights (FWs), revealing improvements in attention, short- and long-term memory ([10]), inhibitory control ([40]; [51]), and general cognitive function ([22]); however, very few have explored the exclusive use of free weights ([11]). With regard to outcome measures, they have frequently been associated with executive functions (EFs) like inhibitory control, working memory, attention, and cognitive flexibility ([14]; [31]). These measures have been assessed using various tests, such as the Montreal Cognitive Assessment (MoCA) ([47]), the Alzheimer’s Disease Assessment Scale–Cognitive Subscale (ADASCog) ([22]), the Wechsler Adult Intelligence Scale (WAIS) ([10]), the Toulouse–Pieron’s Concentration Attention Test ([10]), the Rey–Osterrieth Complex Figure (ROCF) ([10]), the Stroop test ([14]), or the Modified Sternberg Task ([31]).

In this scenario, it may be worthwhile to explore the potential benefits of chronic RT in healthy young adults, considering the use of a methodology that enables the online evaluation of cognitive performance in higher-order cognitive tasks, such as written-text processing, a multifaceted cognitive process that requires sustained attention, working memory, and the integration of information into a coherent mental representation ([34]; [48]). One of these methodologies is eye tracking (ET), a device that allows for the measurement of eye movement, gaze behavior, and pupil dilation. ET allows for assessing performance based on various saccadic movements during information processing, assuming that longer fixation and regression times indicate higher processing costs. Eye movement research in the context of reading is grounded in the eye–mind hypothesis ([33]), which posits a strong relation between eye gaze and cognitive processes and assumes that longer fixation times indicate a higher cognitive load imposed by a task. Studies employing this methodology have uncovered important aspects of the interaction between vision and cognition in the reading process, such as the amount of information gathered during a single fixation, the time required for different types of information extraction from words, and the challenges readers encounter in comprehending textual material ([32]).

Given that at present there is no evidence regarding the impact of RT on the cognitive function of healthy young adults engaged in higher-order processing tasks, the aim of this study is to determine the impact of RT with FWs on written-text processing, utilizing eye-tracking technology. Our methodology relies on the well-established eye–mind hypothesis ([32]), which directly links eye fixations to cognitive processing. Eye movements during reading provide real-time indicators of cognitive processing effort, as fixation duration reflects the cognitive effort required for information processing. Since readers typically fixate on most words in texts ([32]), these fixation patterns offer reliable data about the cognitive demands of text processing. Thus, rather than measuring post hoc comprehension, our study focused on the more immediate and objective measures of cognitive processing through eye tracking, following current trends in reading cognition research ([16]).

## 2. Materials and Methods

### 2.1. Experimental Approach to the Problem

To assess the impact of an RT intervention on text processing, we conducted a clinical trial employing an experimental design including both experimental and control groups. Participants were randomly assigned to either group using concealed allocation, facilitated by Excel’s randomization function. Moreover, the assessor remained blinded throughout the study, ensuring unbiased evaluation. Cognitive function, assessed through ET metrics (including FPRTs and TRTs), was evaluated at the study’s outset and again after 10 weeks of the training intervention in both groups. The experimental group engaged in a 10-week program of free-weight strength training, which encompassed push, pull, and leg exercises. In the second week, the experimental group determined their indirect one-repetition maximum (RM) values as part of a technical learning phase for the exercises and underwent a re-evaluation in the tenth week.

### 2.2. Participants

Twenty-two students from Pontificia Universidad Católica de Valparaíso (PUCV) participated in this study, consisting of seventeen women and five untrained men aged between 19 and 28. All participants were Chilean, native Spanish speakers, and fourth-year students in the Translation and Interpreting program at PUCV. As fourth-year students, all had achieved a certified B2 English proficiency level and corresponding reading skills. Given that they were all in the same year and program, their academic workload and formal reading hours were similar. As exclusion criteria for the subjects, we considered individuals with health issues that could interfere with training, individuals who had participated in any training programs in the six months prior to the study, and those with abnormal or uncorrected vision that could interfere with eye tracking. This information was collected through a registration form completed by the participants themselves (Table A1).

This study is registered at ClinicalTrials.gov (identifier: NCT06662487). Additionally, the study received ethical approval from the Bioethics and Biosafety Committee at PUCV (BIOEPUCV-HB 546-2022, 8th September 2022), and written consent was obtained from each participant. All participants were randomized after signing the consent form to maintain blinding ([46]). Additionally, the evaluator was also blinded to the group to which the participants belonged. This ensured that the study followed a double-blind design ([46]).

This study represents a first step in exploring the impact of free-weight training on written-text processing, a higher-order cognitive measure. As no prior studies directly comparable to ours are available, and no effect sizes were found in the literature, we followed the recommendations of [5] ([5]), who discusses sample size requirements for exercise-based interventions, particularly in studies with strength as a primary outcome, and [21] ([21]), who provide guidance on selecting the appropriate method for sample size calculation based on the statistical test. Following these recommendations, we used an effect size (f) of 0.80, an alpha value of 0.05, and a statistical power of 0.80, configured for a repeated-measure ANOVA with two groups and two time points for evaluation, based on the characteristics of our data. Based on this calculation, 12 participants were required. While this approach has limitations, we consider it a useful starting point for this study and an initial guide for future research in this emerging area. Figure 1 shows the distribution of the participants and the blinding of the study design.

A total of 18 young adult university students completed their participation in this study; overall, significant differences in terms of age were only found between the experimental and control groups. Table 1 shows a description of the baseline values of the participants.

### 2.3. Materials

#### 2.3.1. RT Program

The program lasted 10 weeks, including 2 weeks prior to the intervention, which aimed at learning the exercises and evaluating the maximum strength. The training frequency consisted of 3 sessions per week, with each session comprising 5 exercises ([22]) organized around push, pull, and leg movement patterns. Free weights targeting large muscle groups were utilized for the exercises ([11]). Specific exercises included bench press, close-grip bench press, military press, dumbbell bench press, dumbbell shoulder press, bar bent-over row, T-grip row, close-grip row, meadow row, dumbbell single-arm row, high-bar squat, deadlift bar, Bulgarian squat, dumbbell lunges, and hip thrust. The program incorporated moderate intensities ([11]), ranging from 60% to 80% of the 1RM ([45]), with a focus on progressive overload. Additionally, there was an increase in volume from week 5, reaching 2–3 sets per exercise. Table 2 shows a description of the RT program that was implemented with the participants.

#### 2.3.2. Processing

Six expository texts written in Spanish on general knowledge topics of language, history, and science were utilized (two for each topic). To ensure the suitability of the texts for the readers’ characteristics, they were extracted from school textbooks. The length of the texts ranged between 110 and 112 words. Each text had two versions, based on their syntactic complexity (high/low). Syntactic complexity was operationalized based on dependency locality, which refers to the distance between syntactically related elements. In local conditions (low syntactic complexity), syntactically related elements (subject–verb–object) appeared adjacently in the sentence, as in “The student completed the assignment before the end of the semester.” In non-local conditions (high syntactic complexity), these elements were separated by an adverbial phrase, increasing processing demands, as in “The student, before the end of the semester, completed the assignment.” This manipulation was systematically applied across all experimental texts, maintaining consistent syntactic relationships while varying their locality. The areas of interest where the effects of syntactic complexity were examined correspond to full sentences. This division in terms of syntactic complexity is due to the fact that complex syntactic structures impose greater cognitive demands on the part of the reader, especially in processing and reading comprehension ([35]), which has been supported by different theories (e.g., active filler hypothesis ([13]); dependency locality theory ([24]; [25]); and surprisal ([27])). Table A2 shows a sample of text in both conditions. Additionally, to ensure that participants were reading attentively, each text was followed by a verification question. Texts were distributed in lists that contained three texts each and were randomly assigned to each participant, ensuring that they did not read the same texts before and after the intervention.

### 2.4. Procedures

#### 2.4.1. Recording 1RM Procedures

To measure an indirect 1RM, the experimental group performed a general warm-up that included low-intensity aerobic work and joint mobility exercises of the muscle groups involved ([37]). Subsequently, a specific warm-up was performed with 10 repetitions of the exercises to be performed, with several repetitions in reserve. Finally, the number of repetitions achieved in the last series with the best possible technical execution (less than 10 repetitions) was recorded. The repetitions achieved with each weight were entered into Brzycki’s formula to obtain the 1RM for each exercise. These baseline values were then used to schedule the remaining 8 weeks of training, ranging from 60% to 80% intensities. In the tenth week, indirect 1RM measurements were recorded again to assess strength gains.

#### 2.4.2. Training Session Procedures

Each training session was supervised by two certified physical trainers and included both a warm-up and conditioning phase ([37]). The warm-up consisted of light aerobic activity and joint mobility work aimed at the main muscle groups to be worked on in the session. Three sessions were carried out per week, incorporating different exercises organized by movement patterns. Consequently, each day was composed of five exercises corresponding to specific movement patterns (push–pull and legs).

#### 2.4.3. Eye-Tracking Task

Participants were tested individually and were informed that the task involved reading with eye-tracking equipment. Prior to commencing the experiment, each participant signed an informed consent form. Following this, the eye tracker was set up, and a nine-point calibration screen was carried out for each participant. The participants were instructed to read and comprehend each text, silently, at their own pace, and indicate their readiness to proceed to the (verification) questions by pressing a keyboard button. To prevent the latent learning effect from the first test, before and after the tests the participants read different texts, but care was taken to ensure that they were of the same length in terms of the number of words and syntactic complexity. In addition, in both instances the eye-tracking measurement was performed between 9 and 11 am. Participants were seated 70 cm from the screen, and a chin rest was used to stabilize the head. Eye movements were recorded monocularly by using an EyeLink Portable Duo (SR Research Ltd., Ottawa, ON, Canada) at a sampling frequency of 500 Hz. The stimuli were presented on a 16” ROG Zephyrus M16 notebook with a refresh rate of 100 Hz and a resolution of 1920 × 1080 pixels.

### 2.5. Data Preparation

For strength-associated measures, the data preparation procedure encompassed an initial phase of cleaning, involving the identification and removal of outliers. Missing data were managed to ensure the dataset’s integrity. Subsequently, variables were transformed, with measurements normalized and adjustments applied to meet the necessary statistical assumptions. The data distribution was checked with the Shapiro–Wilk test and with the visual analysis of residuals and Q-Q plots. Following the recommended procedures for data cleaning in reading experiments with eye tracking ([20]), fixations shorter than 80 ms were either merged with a nearby fixation (if the distance between the fixations was <1°) or removed from the data. Two eye movement measures associated with particular patterns of text processing were used: FPRTs, which correspond to the sum of the duration of all fixations on the first pass within an area of interest, and TRTs, the sum of the duration of all fixations that fall within an area of interest ([16]). The reading time measures were skewed and consequently transformed. The best-fitting transformation was selected to normalize the measures; FPRTs were square root transformed and TRTs were logarithmically transformed. A total of 216 eye-tracking observations were analyzed, comprising 108 from before the test, conducted prior to the training, and 108 from after the test, conducted afterward. The dataset included 117 observations from the control group and 99 from the experimental group.

### 2.6. Data Analysis

To observe the expected differences in reading performance between the experimental and control groups after the intervention period, data were analyzed with linear mixed models (LMMs) by using the lme4 package ([4]) in R statistical software (Version 4.0.1; [44]). Several models were constructed, each focusing on a specific eye movement measure corresponding to individual target sentences within the texts. Variables such as moment (pre vs. post), group (control vs. experimental), and syntactic complexity (low vs. high) were incorporated into these models as fixed effects. Random intercepts for both participants and items were included in the models ([1]). The models were constructed with a maximal random structure ([3]). In instances where the full random structure led to convergence issues, a top-down trimming process was applied to the random structure, initially considering correlations between factors ([6]). For two models that failed to converge when using only random intercepts for participants and items, non-significant interaction terms among fixed effects were progressively removed, starting with those associated with the smallest t or z values. Due to difficulties in precisely determining degrees of freedom for statistics estimated by linear mixed models (LMMs), exact *p*-values or degrees of freedom were not ascertainable. Hence, instead of reporting specific *p*-values, statistical significance at the 0.05 level was inferred based on |t or z| values exceeding 1.96 ([1]). Given the observed differences between groups and time, subsequent analysis of variance (ANOVA) tests were conducted to confirm that such differences were due to increased strength. This time, strength was categorized into pre–post values in order to compare whether higher levels of post-intervention strength are associated with improvements in eye-tracker measures. The *p*-value and effect size were computed for each comparison (Hedges’ G). The G of edges was used to adjust the effect size to the sample size.

## 3. Results

To examine the effect of RT on text processing patterns, we conducted a linear mixed model analysis evaluating the effects of group (control vs. experimental), moment (pre vs. post), and syntactic complexity (low vs. high) on FPRTs. The main effect of group was significant, with the experimental group showing higher FPRTs compared to the control group (β = 17.80, SE = 5.38, t = 3.31, CI = [7.19, 28.42]); however, the main effects of moment (β = −1.70, SE = 5.12, t = −0.33, CI = [−11.79, 8.38]) and syntactic complexity (β = 3.12, SE = 5.53, t = 0.56, CI = [−7.79, 14.02]) were not significant. Interaction effects provided further insights into the relationships between factors. The interaction between group and syntactic complexity was significant (β = −27.68, SE = 7.12, t = −3.89, CI = [−41.72, −13.64]), suggesting that the experimental group exhibited reduced FPRTs under high syntactic complexity compared to the control group. Additionally, the three-way interaction among group, moment, and syntactic complexity was significant (β = 25.52, SE = 10.25, t = 2.49, CI = [5.31, 45.73]), highlighting a nuanced relationship among these factors (see Figure 2 and Table A1).

In addition to the linear mixed model analysis, Type III Wald chi-square tests were performed to confirm the significance of main effects and interactions. The main effect of group was significant (χ^2^(1) = 10.93, *p* < 0.001), whereas moment (χ^2^(1) = 0.11, *p* = 0.739) and syntactic complexity (χ^2^(1) = 0.32, *p* = 0.573) were not. A significant interaction was observed between group and syntactic complexity (χ^2^(1) = 15.10, *p* < 0.001), and the three-way interaction among group, moment, and syntactic complexity was also significant (χ^2^(1) = 6.20, *p* = 0.013).

We conducted a second linear mixed model analysis to assess the effects of group (control vs. experimental), moment (pre vs. post), and syntactic complexity (low vs. high) on TRTs. The main effect of the group was significant, with the experimental group showing longer TRTs compared to the control group (β = 0.41, SE = 0.13, t = 3.14, CI = [0.15, 0.67]). In contrast, the main effects of moment (β = −0.10, SE = 0.12, t = −0.86, CI = [−0.34, 0.13]) and syntactic complexity (β = 0.12, SE = 0.19, t = 0.62, CI = [−0.26, 0.50]) were not significant. Interaction effects provided additional insights into how these factors interacted. A significant group in terms of syntactic complexity interaction (β = −0.36, SE = 0.16, t = −2.26, CI = [−0.68, −0.05]) suggested that participants in the experimental group demonstrated reduced TRTs under conditions of high syntactic complexity compared to the control group; however, the three-way interaction among group, moment, and syntactic complexity was not significant (β = 0.35, SE = 0.23, t = 1.49, CI = [−0.11, 0.81]), indicating no additional complexity in the relationship among these factors (see Figure 3 and Table A3).

To complement the mixed model results, Type III Wald chi-square tests were conducted to evaluate the significance of the main effects and interactions. The main effect of group was confirmed to be significant (χ^2^(1) = 9.85, *p* = 0.002), whereas moment (χ^2^(1) = 0.75, *p* = 0.387) and syntactic complexity (χ^2^(1) = 0.38, *p* = 0.538) were not. The interaction between group and syntactic complexity was significant (χ^2^(1) = 5.12, *p* = 0.024), further supporting the reduction in reading times for the experimental group under high syntactic complexity; however, the three-way interaction among group, moment, and syntactic complexity did not reach significance (χ^2^(1) = 2.22, *p* = 0.137).

Given our aim to investigate the effect of strength training on participants’ cognitive performance, we conducted additional analyses in the experimental group. Specifically, ANOVAs were conducted according to the average strength levels of the pre–post-intervention maximal tests (mean pre = 31.5 ± 9.50; mean post = 42.7 ± 11.1). Figure 4 shows the comparison of the eye-tracker measurements obtained with the pre–post-intervention strength levels globally and separated by syntactic complexity. Overall, in TRTs and FPRTs we found significant differences attributed to the increase in strength after the intervention. In particular, there were significant differences in favor of increased strength in overall reading times and at low syntactic complexity. Similarly, in FPRTs we found significant differences in high and low syntactic complexity.

## 4. Discussion

Our analysis of eye movement data revealed distinct patterns of interaction between RT and text processing. For FPRTs, the linear mixed model analysis showed a significant main effect of group, with the experimental group exhibiting higher FPRTs compared to the control group. Neither moment nor syntactic complexity showed significant main effects. The control group maintained consistent processing patterns across pre and post measurements, showing longer processing times in high-complexity conditions compared to low-complexity conditions. Interestingly, the experimental group demonstrated an inverse pattern. Prior to the intervention, they showed longer processing times for low-complexity compared to high-complexity conditions; however, post-intervention, this pattern reversed, with increased processing times in high-complexity conditions. This shift suggests enhanced attention to complex syntactic structures following the training period, particularly as the effect manifests in early processing measures (FPRTs), indicating increased processing focus. Notably, the three-way interaction between group, moment, and syntactic complexity in FPRTs provided compelling evidence of training-induced effects on cognitive processing.

Regarding TRTs, while not showing a significant three-way interaction, the analysis revealed the experimental group’s general trend toward decreased reading times between pre and post measurements, particularly in low-complexity conditions, suggesting improved processing efficiency in conditions of no syntactic difficulties, following the resistance training intervention.

Given our aim to investigate the effect of strength training on participants’ cognitive performance, we conducted additional analyses in the experimental group. Specifically, ANOVAs were performed according to the average strength levels of the pre–post-intervention maximal tests. The data revealed decreased reading times post-intervention across syntactic complexity levels, suggesting a general improvement in processing efficiency associated with increased strength. Notably, FPRTs increased significantly post-intervention under high-complexity conditions, indicating enhanced attention allocation to complex structures. Additionally, in low-complexity conditions, both TRTs and FPRTs decreased, demonstrating improved processing efficiency for simpler syntactic structures.

One of the findings that encourages further exploration of this relationship is the contrast in FPRTs within the experimental group under the high-syntactic-complexity condition. This condition inherently imposes a higher cognitive load due to the intricate nature of the syntactic structures, which typically require greater cognitive resources and lead to extended reading times compared to low-complexity conditions. While we expected RT to reduce processing times across all complexity conditions, our findings revealed an unexpected pattern: increased early processing times (FPRTs) specifically for high-complexity texts in the experimental group post-intervention. Although seemingly counterintuitive, this pattern likely reflects complex interactions between cognitive mechanisms and training intervention effects.

The increased focus and attentional demands inherent in RT with free weights (FWs) may have selectively enhanced participants’ capacity to allocate attentional resources to more-demanding cognitive tasks, such as processing complex syntactic structures. This improvement in high-complexity conditions during early processing suggests that the intervention might have facilitated more effective engagement with intricate structures, potentially mediated by enhancements in EFs like working memory and attentional control. These findings align with prior research indicating that strength training can improve selective attention and inhibitory control, both of which are critical for managing cognitive load during challenging tasks ([40]; [51]). The unexpected augmentation of fixation durations following the intervention challenges the expectation of decreased reading times, presumed due to the cognitive advantages gained from strength training. This deviation prompts a critical inquiry into the relationship between strength training and cognitive function, particularly their intricate interplay with discourse processing.

The effects of apparently increased focus on processing texts with high syntactic complexity, reflected in the increased FPRTs found in both the mixed model analysis and ANOVA, along with the decreased reading times of the experimental group in TRTs and FPRTs for texts with low syntactic complexity (ANOVA), could be explained by the positive effects of RT on EFs. Regarding the relationship between RT and EFs, [40] ([40])’s study, conducted on older adults, showed positive effects on inhibitory control as a result of a chronic RT intervention; however, our results may be more directly associated with findings in young adults, such as those obtained by [51] ([51]), which showed that improvements in inhibitory control are more pronounced in RT with FWs interventions compared to machine-based RT, with the caveat that this study measured acute effects. Another EFs that improves as a result of RT, which could potentially enhance reading and writing performance, is memory, as evidenced by [10] ([10]), who reported significant improvements in both short-term and long-term memory in healthy older adults. The same was observed in the study by [40] ([40]), which revealed enhancements in associative memory.

Regarding the relationship between EFs and reading performance, it has been observed ([42]; [41]) that EFs plays a fundamental role in processing and comprehending written texts by orchestrating specific processes, such as information integration, retrieval from the mental lexicon, strategy utilization, and simultaneous engagement in multiple reading tasks. Studies have also demonstrated that updating working memory aids processing by maintaining the activation of relevant information during reading; inhibitory control aids processing by restricting the activation of irrelevant text details and preventing unwanted memory intrusions; and shifting attention supports processing by integrating different types of information and focusing on various text features and situational contexts ([23]). Therefore, the potential impact of RT on EFs could explain why participants in the experimental group demonstrate greater efficiency as readers, particularly in terms of TRTs and FPRTs under low-syntactic-complexity conditions (ANOVA). Thus, the potential impact of RT on EFs could be related to improvements in reading ability.

Another possible explanation for the positive effects of increased focus on texts with high syntactic complexity and shorter TRTs and FPRTs in low syntactic complexity as a result of RT with FWs refers to the fact that this type of intervention selectively improves aspects of cognition due to differential demands. For example, individuals participating in RT with FWs need to pay constant attention to what they are doing to avoid injuring themselves or those around them. In this sense, these periods of vigilance could act as a form of attentional training and explain why there was an improvement in performance on EFs tests, as many of these tasks assess an individual’s ability to attend to specific stimuli ([36]). Although significant differences between the effects of RT with FWs and machine-based lifting on athletic performance and muscular architecture have not been found ([28]), it is evident that attentional demands differ between these two modalities of RT. Exercises using machines isolate muscles and follow predetermined paths; in contrast, FW exercises used in our intervention require individuals to pay attention to a greater number of variables related to coordination and balance to maintain control over the movement of the weights ([37]). These more complex execution conditions could impact attentional improvement, potentially explaining the effects of the intervention on reading times; however, it would be interesting to evaluate this hypothesis in future studies that directly compare these two forms of RT.

From another perspective, it is plausible that the effects of RT on EFs, and consequently its positive impact on text processing, may be mediated by neurobiological mechanisms unrelated to the specific cognitive demands of the exercise. These mechanisms include increases in molecules such as brain-derived neurotrophic factor (BDNF) and proteins like insulin-like growth factor 1 (IGF-1), which are associated with exercise’s effects on learning and depression, as well as the combined action of IGF-1 and vascular endothelial growth factor (VEGF) on hippocampal angiogenesis and neurogenesis ([7]). These molecular changes are thought to induce structural alterations, such as increased gray and white matter volume ([36]), which could also lead to cognitive changes, such as an impact on learning and memory, considering the highly plastic nature of the hippocampus ([9]). It is also worth noting that exercise acts as a stimulus for new blood vessel formation, and increased cerebral blood flow correlates with cognitive enhancement ([10]; [8]). In fact, [8] ([8]) demonstrated that physical fitness was positively associated with the number of small blood vessels in older individuals undergoing magnetic resonance angiography, indicating angiogenesis, a phenomenon not observed in sedentary individuals.

It is also possible that neurobiological and cognitive mechanisms work synergistically. For instance, neurobiological mechanisms may enhance neuroplasticity ([18]; [9]), which in turn might have an impact on EFs that are more consistently engaged during resistance exercises, such as attention ([36]), inhibitory control ([42]), and associative memory ([40]); however, clear connections between exercise, neurobiological mechanisms, and cognitive changes still need further investigation.

## 5. Conclusions

RT with FWs showed interesting results, considering the increase in FPRTs for texts with high complexity, which suggests a greater capacity for focus on more intricate texts and a decrease in TRTs and FPRTs for texts with lower complexity. These effects found in the various analyses could be explained by the proven improvements that RT generates in EFs, such as attention, inhibitory control, and memory. These, in turn, relate to improvements in text processing. From a neurobiological perspective, these positive effects could also stem from molecular adaptations and neuroplasticity, elements that should be considered and measured in future research efforts. It can be concluded that RT with free weights has a positive effect on text processing; however, it would be interesting to make a direct comparison between RT with machines to glimpse possible differences. Additionally, there is a small number of studies on the effects of RT on cognitive function in young adults, and even fewer on variables measured through eye tracking. Therefore, this scarcity opens a line of research in which these variables could even intersect with EFs, thus understanding RT as an enhancer of cognitive and academic performance in this age group.

## Figures and Tables

**Figure 1 behavsci-15-00077-f001:**
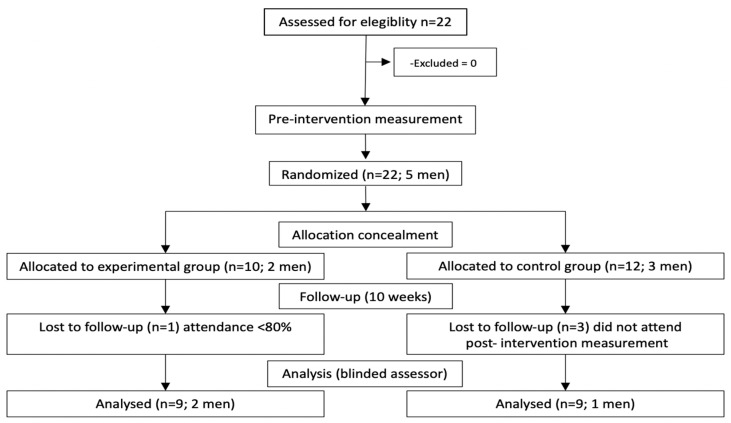
Consort flow chart of the study.

**Figure 2 behavsci-15-00077-f002:**
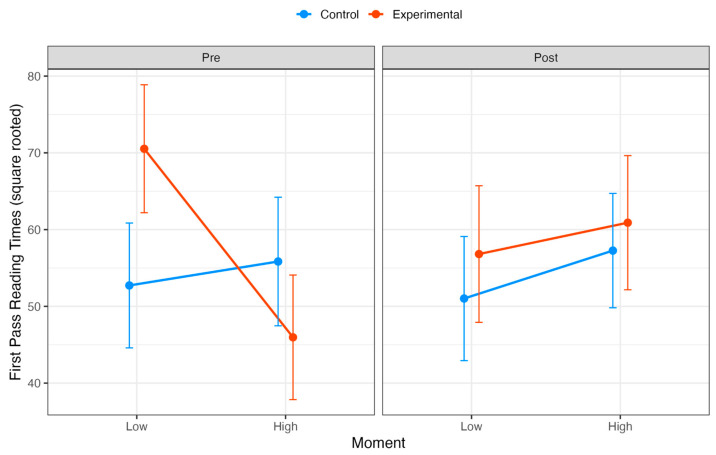
FPRTs (square-rooted) for the control (blue line) and experimental (red line) groups across two moments (pre and post) and two levels of syntactic complexity (low and high). Error bars represent 95% confidence intervals for the mean estimates.

**Figure 3 behavsci-15-00077-f003:**
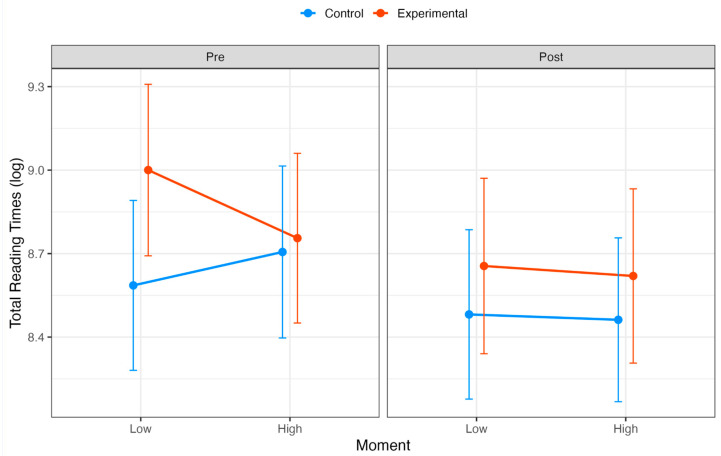
TRTs (log-transformed) for the control (blue line) and experimental (red line) groups across two moments (pre and post) and two levels of syntactic complexity (low and high). Error bars represent 95% confidence intervals for the mean estimates.

**Figure 4 behavsci-15-00077-f004:**
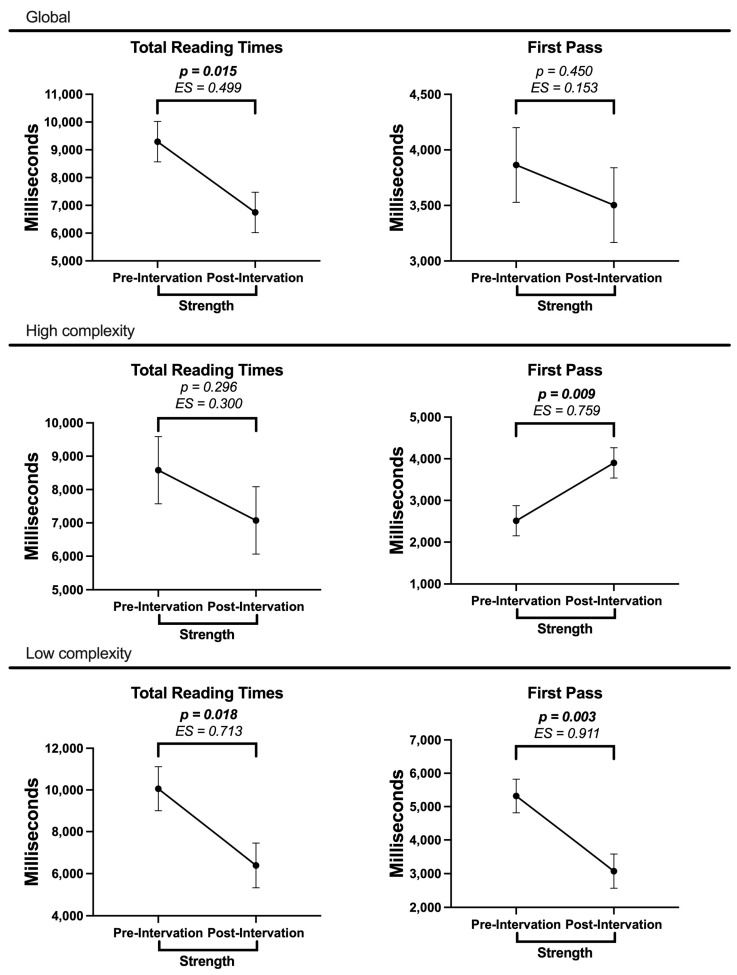
ANOVA results between pre–post-intervention strength values and eye-tracker measurements. In bold, significant *p*-values (<0.05) are presented. ES = effect size (Hedges’ g).

**Table 1 behavsci-15-00077-t001:** Participants’ baseline values. *p*-value of *t*-test for independent samples for continuous variables and chi-square for category variables. (cm): centimeters; (kg): kilograms. *p*-value < 0.005 in bold.

Variable	All (n = 18)	Experimental (n = 9)	Control (n = 9)	*p*-Value
Sex				0.555
Women	15 (83.3%)	7 (38.9%)	8 (44.4%)	
Men	3 (16.7%)	2 (11.1%)	1 (5.6%)	
Age (years)	21.0 ± 2.14	20.0 ± 0.86	22.0 ± 2.60	**0.044**
Height (cm)	161 ± 0.07	1.62 ± 0.09	1.59 ± 0.04	0.340
Weight (kg)	58.9 ± 10.3	62.3 ± 12.8	55.4 ± 5.66	0.159
BMI	22.8 ± 3.49	23.6 ± 4.11	22.0 ± 2.75	0.338

**Table 2 behavsci-15-00077-t002:** Resistance training (RT) program.

Exercise	Week 1	Week 2	Week 3	Week 4	Week 5	Week 6	Week 7	Week 8
Day 1								
High-bar squat	Int: 60% RM	Int: 65% RM	Int: 70% RM	Int: 75% RM	Int: 70% RM	Int: 75% RM	Int: 80% RM	
Bar deadlift	Rep: 12	Rep: 10	Rep: 8	Rep: 6	Rep: 8	Rep: 6	Rep: 4	Control
Bulgarian squat	S: 2	S: 2	S: 2	S: 2	S: 3	S: 3	S: 3	1RM
Dumbbell lunge	R: 2′	R: 2′	R: 2′	R: 2′	R: 2′	R: 2′	R: 2′	
Hip thrust								
Day 2								
Bench press	Int: 60% RM	Int: 65% RM	Int: 70% RM	Int: 75% RM	Int: 70% RM	Int: 75% RM	Int: 80% RM	
Close-grip bench press	Rep: 12	Rep: 10	Rep: 8	Rep: 6	Rep: 8	Rep: 6	Rep: 4	Control
Military press	S: 2	S: 2	S: 2	S: 2	S: 3	S: 3	S: 3	1RM
Dumbbell bench press	R: 2′	R: 2′	R: 2′	R: 2′	R: 2′	R: 2′	R: 2′	
Dumbbell shoulder press								
Day 3								
Bent-over bar row	Int: 60% RM	Int: 65% RM	Int: 70% RM	Int: 75% RM	Int: 70% RM	Int: 75% RM	Int: 80% RM	
T-grip row	Rep: 12	Rep: 10	Rep: 8	Rep: 6	Rep: 8	Rep: 6	Rep: 4	Control
Close-grip row	S: 2	S: 2	S: 2	S: 2	S: 3	S: 3	S: 3	1RM
Meadow row	R: 2′	R: 2′	R: 2′	R: 2′	R: 2′	R: 2′	R: 2′	
Dumbbell single-arm row								

Int: intensity; Rep: repetition; S: series; R: rest; and 1RM: one-repetition maximum.

## Data Availability

The data presented in this study are available upon request from the corresponding author.

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
