# Peer review of "The Effect of Free Weight Resistance Training on Cognitive Function Explored Through Eye Tracking: A Randomized Double-Blind Clinical Trial"

_behavsci, 2025, doi:10.3390/bs15010077_

Round 1
Reviewer 1 Report
Comments and Suggestions for Authors
This article attempts to discuss the impact of RT on reading, but it lacks sufficient description and analysis of key tasks.
1. In the entire abstract, only the last sentence "The results show significant improvements in eye-tracking measures: Total Reading Times and First Pass Reading Times. These improvements can be attributed to RT, as it has been shown to enhance executive functions, which in turn influence reading performance" introduces the results. The rest of the abstract mainly elaborates on the research objectives and methods, making the structure incoherent. Even so, based on the results mentioned above, it is not clear where the innovation of this study lies and how the differences between the two groups are demonstrated.
2. "Our interest is unveiling the potential benefits of RT on cognitive function among healthy young adults engaged in complex cognitive tasks.” Why is it stated that the task in this study is a complex cognitive task? Text reading is a common task in eye tracking research, but where does its complexity lie?"
3. "considering the use of a methodology that enables the on-line evaluation of cognitive performance in higher-order processing tasks, such as written text processing”. This description does not reference any literature and does not explain why written text processing is considered a higher-order processing task.
4. The article does not explain how to determine if the participants truly understand the material they have read. Without assessments, it is not possible to infer whether the participants comprehend the material based on differences in eye movements. If understanding is not sufficient, what can differences in eye movements indicate?
5. The effect used when determining the sample size may not necessarily be appropriate, it is recommended to refer to relevant literature and recalculate the sample size.
Author Response
We extend our sincere gratitude to the reviewers for their meticulous and insightful comments, greatly enhancing the quality of our study. In response to their invaluable suggestions, we have integrated additional information into the manuscript, highlighted in yellow to underscore the implemented modifications.
- In the entire abstract, only the last sentence "The results show significant improvements in eye-tracking measures: Total Reading Times and First Pass Reading Times. These improvements can be attributed to RT, as it has been shown to enhance executive functions, which in turn influence reading performance" introduces the results. The rest of the abstract mainly elaborates on the research objectives and methods, making the structure incoherent. Even so, based on the results mentioned above, it is not clear where the innovation of this study lies and how the differences between the two groups are demonstrated.
R: Thank you for your valuable feedback on our abstract's structure. We have revised it to better highlight our findings, for both Total Reading Times and First Pass Reading Times, and detailed outcomes across different complexity conditions. These modifications better demonstrate the differences between groups while maintaining scientific rigor. Lines 14 to 25
- "Our interest is unveiling the potential benefits of RT on cognitive function among healthy young adults engaged in complex cognitive tasks.” Why is it stated that the task in this study is a complex cognitive task? Text reading is a common task in eye tracking research, but where does its complexity lie?"
R: Thank you for your insightful observation about the complexity of reading as a cognitive process. We have addressed this by adding an explanation and supporting references on page 2 lines 64 to 69, specifically discussing how text processing involves multiple cognitive processes including information integration.
- "considering the use of a methodology that enables the on-line evaluation of cognitive performance in higher-order processing tasks, such as written text processing”. This description does not reference any literature and does not explain why written text processing is considered a higher-order processing task.
R: Thank you for your insightful observation cognitive process. We have addressed this by adding an explanation and supporting references on page 2 lines 64 to 69, specifically discussing how text processing involves multiple cognitive processes and is consider as a higher-order cognitive task.
- The article does not explain how to determine if the participants truly understand the material they have read. Without assessments, it is not possible to infer whether the participants comprehend the material based on differences in eye movements. If understanding is not sufficient, what can differences in eye movements indicate?
R: Thank you for your important question about comprehension assessment. We have addressed this concern by adding a detailed explanation on page 2 lines 85 to 93, that justifies our focus on processing measures through the eye-mind hypothesis (Just & Carpenter, 1980). This addition explains how eye-tracking provides real-time, objective indicators of cognitive processing during reading, supporting our methodological approach.
- The effect used when determining the sample size may not necessarily be appropriate, it is recommended to refer to relevant literature and recalculate the sample size.
R: Thank you very much for your comment regarding the sample size calculation. We acknowledge that our study represents a first step in investigating the impact of free-weight training on written text processing, a higher-order cognitive measure. Since no directly comparable prior studies or effect sizes are available in the literature, we relied on the recommendations of Beck (29), who discusses sample size considerations for exercise-based interventions, particularly in studies with strength as a primary outcome, and Faul et al (30), who provide a guide for selecting the appropriate method for sample size calculation based on the statistical test. Following these recommendations, we used an effect size (f) of 0.80, an alpha value of 0.05, and a statistical power of 0.80, configured for a repeated measures ANOVA with two groups and two time points for evaluation, based on the characteristics of our data. While we recognize the limitations of this approach, we consider it a useful starting point for this exploratory study and an initial guide for future research in this emerging area. We have revised the manuscript to include this information more clearly lines 127 to 138.
Reviewer 2 Report
Comments and Suggestions for Authors
Dear,
First of all, I would like to congratulate the authors on their motivation to write this manuscript. The title of the paper, The effect of Free Weight Resistance Training on Cognitive Function Explored through Eye Tracking: A Randomized Double-Blind Clinical Trial, suggests that the study focuses on the experimental theme of determining how training/exercise affects the development of cognitive abilities. Neuromuscular function is one of the most important and decisive factors and goals in resistance training. Previous research indicates that resistance training, in addition to its physical benefits, helps in preventing a decline in cognitive abilities. Furthermore, resistance training affects the development of brain functions and increases the production of brain-derived neurotrophic factor, which is involved in neuroplasticity and learning. The goal of this study is to investigate the impact of resistance training, specifically with free weights, on cognitive function among healthy young adults using Eye Tracking technology. The study involved 22 participants aged 19-28 years. Participants were divided into a control and an experimental group. The experimental group participated in a 10-week free weight strength training program. Cognitive function was operationalized as written text processing and assessed using Eye Tracking metrics. The results showed significant improvements in eye-tracking measures: total reading time and first-pass reading time. These improvements can be attributed to resistance training, as it has been shown to enhance executive functions, which in turn affect reading performance.
The manuscript submitted for review features a well-balanced structure of information. The author describes the research in a logical and consistent manner, the research results are presented in an accessible way, and the conclusions drawn from the results are aligned accordingly. The introduction is a well-written theoretical part that continues with evidence from previous studies. The aim of the paper is clearly stated, and a hypothesis could possibly be added. The methods are explained in detail, from the procedure of sample collection and inclusion of participants to the sample size calculation. However, the exclusion criteria for participants were not established. The experimental program and methods of data collection and processing are clearly described. The use of images and tables allows for a visual representation of the results, making them easier to analyze. The results are consistent with the research methods used. The discussion presents the key findings of the study, and the authors compare them with previous research. At the end of the discussion, the study's strengths are highlighted, and recommendations for future research of this type are provided.
The manuscript is well-structured and relevant to the research field. The literature used is mostly recent.
Finally, I believe this article is acceptable for publication in the journal, and my suggestion is to accept the manuscript.
Author Response
We extend our sincere gratitude to the reviewers for their meticulous and insightful comments, greatly enhancing the quality of our study. In response to their invaluable suggestions, we have integrated additional information into the manuscript, highlighted in yellow to underscore the implemented modifications.
- First of all, I would like to congratulate the authors on their motivation to write this manuscript. The title of the paper, The effect of Free Weight Resistance Training on Cognitive Function Explored through Eye Tracking: A Randomized Double-Blind Clinical Trial, suggests that the study focuses on the experimental theme of determining how training/exercise affects the development of cognitive abilities. Neuromuscular function is one of the most important and decisive factors and goals in resistance training. Previous research indicates that resistance training, in addition to its physical benefits, helps in preventing a decline in cognitive abilities. Furthermore, resistance training affects the development of brain functions and increases the production of brain-derived neurotrophic factor, which is involved in neuroplasticity and learning. The goal of this study is to investigate the impact of resistance training, specifically with free weights, on cognitive function among healthy young adults using Eye Tracking technology. The study involved 22 participants aged 19-28 years. Participants were divided into a control and an experimental group. The experimental group participated in a 10-week free weight strength training program. Cognitive function was operationalized as written text processing and assessed using Eye Tracking metrics. The results showed significant improvements in eye-tracking measures: total reading time and first-pass reading time. These improvements can be attributed to resistance training, as it has been shown to enhance executive functions, which in turn affect reading performance.
R: Thank you for your assessment of the contribution of our study and the comprehensive summary you provided.
- The manuscript submitted for review features a well-balanced structure of information. The author describes the research in a logical and consistent manner, the research results are presented in an accessible way, and the conclusions drawn from the results are aligned accordingly. The introduction is a well-written theoretical part that continues with evidence from previous studies. The aim of the paper is clearly stated, and a hypothesis could possibly be added. The methods are explained in detail, from the procedure of sample collection and inclusion of participants to the sample size calculation. However, the exclusion criteria for participants were not established. The experimental program and methods of data collection and processing are clearly described. The use of images and tables allows for a visual representation of the results, making them easier to analyze. The results are consistent with the research methods used. The discussion presents the key findings of the study, and the authors compare them with previous research. At the end of the discussion, the study's strengths are highlighted, and recommendations for future research of this type are provided.
R: Thank you for your detailed assessment of our study, as well as for recognizing its coherence and organized structure. Regarding the exclusion criteria, we have clarity on this matter, but we appreciate your observation and are committed to specifying more clearly in the text that we exclude from the study those individuals who have visual problems that prevent eye tracking with the eye tracker, as well as subjects with incompatible physical health to participate in the training program and those who have participated in any strength training program in the six months prior to the study. Lines 115 to 119.
- The manuscript is well-structured and relevant to the research field. The literature used is mostly recent.
R: Thank you for recognizing that our study is well-structured and for valuing and acknowledging that the majority of our references are up to date
- Finally, I believe this article is acceptable for publication in the journal, and my suggestion is to accept the manuscript.
R: We appreciate your comments and your recommendation for our study to be published.
Reviewer 3 Report
Comments and Suggestions for Authors
The study addresses a novel research question as it seeks to find whether free weight fitness training improves cognitive performance in healthy young adults. The researchers assigned participants randomly to two different fitness regimen groups for the intervention and recorded participants eye-movements while they read written Spanish texts of varied syntactic complexity to capture participants cognitive performance.
I commend the authors on using a novel technique to get fitness benefits assessments, but I am afraid I cannot recommend the ms for publication with Behavioral Sciences journal in its current form as the results do not seem to support the authors conclusions. I offer a few suggestions on what aspects of the project need more expansions/revisions, but I will leave it to the editor to decide whether these revisions seem feasible and enough to salvage the project.
Major issues:
1: control and experimental groups seem to be differing significantly by age with experimental group being a younger one. To bypass this confound did the authors get any independent measures of language knowledge like vocabulary or reading comprehension? These measures, if collected might help alleviate the group differences induced by age if clearly shown that language and reading comprehension did not differ in the groups pre-test. If the eye-tracking stimuli all were followed up by comprehension questions then the accuracy need to be reported as this might help alleviate concerns for the between group homogeneity.
2: It is very puzzling and not intuitive to see that in the experimental group the more complex syntax condition was generally easier than the simple syntax condition. Why is that? It seems that the experiment was not doing what it was designed to do. At a minimum, as a sanity check the complex syntax should be processed slower and it does not seem to be the case here.
3. I only saw models within control and experimental group but there seem to be a model missing that has group as a variable and directly compares pre- and post-test performance across simple and complex syntax between control and experimental groups: First pass (or Total Time) ~ Group*Moment*Syntax. Without this model we cannot draw a direct conclusion regarding group performance differences.
Points that need more explanations/motivations:
· How the syntactic complexity was defined? I would need to see a better rationale and description of the syntactic manipulations. Was it always the subject and the verb were split by the adverbial modifying phrase as in the example while in the simple syntax the subject and the predicate were next to each other? If this is the case, have the authors examined first pass and total time reading measures on the subject and the predicate across conditions?
· How interest areas where defined – when the authors say first pass and total time do they mean sentence or whole passage (i.e.trial)?
· How many total observations were included in the analyses?
Author Response
We extend our sincere gratitude to the reviewers for their meticulous and insightful comments, greatly enhancing the quality of our study. In response to their invaluable suggestions, we have integrated additional information into the manuscript, highlighted in yellow to underscore the implemented modifications.
- Control and experimental groups seem to be differing significantly by age with experimental group being a younger one. To bypass this confound did the authors get any independent measures of language knowledge like vocabulary or reading comprehension? These measures, if collected might help alleviate the group differences induced by age if clearly shown that language and reading comprehension did not differ in the groups pre-test. If the eye-tracking stimuli all were followed up by comprehension questions then the accuracy need to be reported as this might help alleviate concerns for the between group homogeneity.
R: Thank you for noting the age differences. All participants were fourth-year Translation and Interpreting students with certified B2 English proficiency and equivalent reading skills. At this academic level, they share comparable reading abilities. A clarification about this was added in lines 110-114. Additionally, age-related reading skill variations are primarily significant during early acquisition years, not among young adults with similar educational backgrounds.
Reference: Oakhill, J., Cain, K., Elbro, C. (2019). Reading Comprehension and Reading Comprehension Difficulties. In: Kilpatrick, D., Joshi, R., Wagner, R. (eds) Reading Development and Difficulties. Springer, Cham. https://doi.org/10.1007/978-3-030-26550-2_5
- It is very puzzling and not intuitive to see that in the experimental group the more complex syntax condition was generally easier than the simple syntax condition. Why is that? It seems that the experiment was not doing what it was designed to do. At a minimum, as a sanity check the complex syntax should be processed slower and it does not seem to be the case here.
R: Thank you for raising this important observation. We agree that the finding appears counterintuitive at first glance. To address this, we have included an extended discussion in the manuscript to explore potential explanations for why, within the experimental group, high syntactic complexity texts were processed faster than low syntactic complexity texts in certain instances (lines 419 to 438). This unexpected result may reflect the impact of the free-weight resistance training (FW RT) intervention on cognitive mechanisms such as attentional allocation and executive function.
- I only saw models within control and experimental group but there seem to be a model missing that has group as a variable and directly compares pre- and post-test performance across simple and complex syntax between control and experimental groups: First pass (or Total Time) ~ Group*Moment*Syntax. Without this model we cannot draw a direct conclusion regarding group performance differences.
R: We sincerely appreciate your insightful comment regarding the need for a model that directly evaluates the interaction between group (Control vs. Experimental), moment (Pre vs. Post), and syntactic complexity (Low vs. High). In response, we have conducted a linear mixed model analysis that incorporates group as a variable alongside moment and syntactic complexity. This model allowed us to assess not only the main effects but also the interaction effects, including the three-way interaction among these variables. The results are now presented in the revised manuscript, highlighting significant findings for the main effect of group, the interaction between group and syntactic complexity, and the three-way interaction among group, moment, and syntactic complexity. Additionally, we have included Type III Wald chi-square tests to further clarify these relationships and provide a comprehensive exploration of the differences across conditions. Figures 2 and 3 have been updated accordingly to reflect these findings. This information was added in lines 280 to 329. Thank you for your valuable suggestion, which has greatly enhanced the robustness of our analysis and the clarity of our results.
- How the syntactic complexity was defined? I would need to see a better rationale and description of the syntactic manipulations. Was it always the subject and the verb were split by the adverbial modifying phrase as in the example while in the simple syntax the subject and the predicate were next to each other? If this is the case, have the authors examined first pass and total time reading measures on the subject and the predicate across conditions?
R: Thank you for your questions about syntactic complexity. We have addressed the definition and detailed description of syntactic manipulations on page 6 lines 182 to 192. Yes, we examined both first pass and total time reading measures on the subject and predicate across conditions.
- How interest areas where defined – when the authors say first pass and total time do they mean sentence or whole passage (i.e.trial)?
R: Thank you for your valuable observation. To address this point, we revised the Materials and Methods section to clarify how the areas of interest (AOIs) were defined (lines 189 to 192) and at what level the eye-tracking metrics were calculated. Specifically, we explained that the AOIs correspond to individual sentences within the passages, not the entire passage or trial.
- How many total observations were included in the analyses?
R: Thank you very much for your comment regarding the number of observations. A total of 216 eye tracking observations were analyzed, comprising 108 from the pre-test, conducted prior to the training, and 108 from the post-test, conducted afterward. The dataset included 117 observations from the control group and 99 from the experimental group. This information was added in lines 249 to 252.
Round 2
Reviewer 1 Report
Comments and Suggestions for Authors
Although the article has made appropriate modifications to previously raised issues, there are still certain problems that exist.
1. The article included content on prior testing for sample size, but it did not specify whether the number of participants in this study meets the basic requirements.
2. RM did not provide the full name when it first appeared.
3. Is the division based on their syntactic complexity (high/low) reasonable or not? Specific explanations or evidence, such as literature, would be appreciated.
4. The text proposes the following idea: "For the Eye Tracking measures, fixations shorter than 80 ms were either merged with a nearby fixation (if the distance between the fixations was < 1°) or removed from the data." My question is, which specific method was employed? Or was it handled arbitrarily? Why?
5. In the conclusion, it is suggested to avoid referencing too much literature. If comparisons with existing literature are necessary, they can be included in the discussion.
Author Response
We extend our sincere gratitude to the reviewers for their meticulous and insightful comments, greatly enhancing the quality of our study. In response to their invaluable suggestions, we have integrated additional information into the manuscript, highlighted in yellow to underscore the implemented modifications.
- The article included content on prior testing for sample size, but it did not specify whether the number of participants in this study meets the basic requirements.
R: Thank you very much for your comment regarding the sample size calculation. Based on the sample size calculation, 12 participants were required. To clarify this, it was added on line 136.
- RM did not provide the full name when it first appeared.
R: Thank you very much for your correction on this matter. RM refers to maximum repetition and was added on line 105.
- Is the division based on their syntactic complexity (high/low) reasonable or not? Specific explanations or evidence, such as literature, would be appreciated.
R: We thank the reviewer for raising this important question regarding the division based on syntactic complexity (SC). Elevated levels of SC, characterized by longer sentences, extensive use of subordinate clauses, and intricate syntactic constructions, are well-documented to impose higher cognitive demands on readers. The division into high and low SC was carefully designed to reflect these established differences in processing demands. High SC texts included sentences with multiple layers of subordination and intricate clause embedding, while low SC texts featured shorter, simpler sentences with fewer subordinate clauses. This categorization aligns with prior research, which has demonstrated that such distinctions can significantly impact reading comprehension and processing, especially in tasks requiring the integration of complex syntactic relationships (Kleijn, 2018). Indeed, there are different theories explaining why certain syntactic structures are more difficult to understand than others, as mentioned in the document on page 6, lines 192 to 197.
We hope this explanation, supported by the cited literature, adequately addresses the reviewer’s concern and justifies the methodological approach employed in our study. If further elaboration is needed, we would be happy to provide additional details.
- The text proposes the following idea: "For the Eye Tracking measures, fixations shorter than 80 ms were either merged with a nearby fixation (if the distance between the fixations was < 1°) or removed from the data." My question is, which specific method was employed? Or was it handled arbitrarily? Why?
R: We appreciate the reviewer’s comments and the opportunity to clarify our methodological approach. The method employed for handling short fixations (<80 ms)—merging them with a nearby fixation if the angular distance is less than 1° or removing them otherwise—is not arbitrary. It follows established practices in the field of eye-tracking research and is supported by empirical findings in the literature.
Eskenazi et al. (2023) provide a comprehensive review, noting that fixations shorter than 60-70 ms often fail to produce meaningful input to the visual system and are typically considered artifacts or noise. These brief fixations lack the duration necessary for effective visual processing, making their removal or integration a standard practice in data cleaning. To preserve meaningful fixations, those shorter than 80 ms were merged with nearby fixations when the angular distance was less than 1°, a threshold commonly used in eye-tracking studies due to the typical precision of eye-tracking systems, which ranges between 0.5° and 1.0° depending on calibration and system characteristics.
This procedure ensures that the processed data reflect genuine visual behavior while minimizing the influence of spurious measurements. By adhering to these established criteria, we ensured the validity and reliability of our eye-tracking data, aligning with best practices in the field as highlighted in Eskenazi et al. (2023). We hope this explanation adequately addresses the reviewer’s concerns, and we remain available for further clarification if needed (lines 241 to 244).
- In the conclusion, it is suggested to avoid referencing too much literature. If comparisons with existing literature are necessary, they can be included in the discussion.
R: We appreciate the reviewer’s comments, we accepted the suggestion’s reviewer and the references were deleted.
Reviewer 3 Report
Comments and Suggestions for Authors
I thank the authors for taking their time to address my questions. I recommend the manuscript for the publication with Behavioral Sciences.
Author Response
We extend our sincere gratitude to the reviewers for their meticulous and insightful comments, greatly enhancing the quality of our study. In response to their invaluable suggestions, we have integrated additional information into the manuscript, highlighted in yellow to underscore the implemented modifications.
- I thank the authors for taking their time to address my questions. I recommend the manuscript for the publication with Behavioral Sciences.
R: We appreciate your comments and your recommendation for our study to be published.